# Deep Learning Regression of Cardiac Phase on Real-Time MRI

**Samira Masoudi**                                              SMASOUDI@UCSD.EDU
**Amin Mahmoodi**                                         AMAHMOODI@HEALTH.UCSD.EDU
**Hafsa S. Babar**                                             HBABAR@HEALTH.UCSD.EDU
**Albert Hsiao**                                               ALHSIAO@HEALTH.UCSD.EDU
*University of California San Diego, San Diego, CA, USA*

## Abstract

Cine steady-state free-precession (SSFP) is the backbone of cardiac MRI, providing visualization of cardiac structure and function over the cardiac cycle, but requires concurrent ECG-gating to combine k-space data over multiple heart beats. However, cine SSFP is limited by a number of factors including arrhythmia, where beat-to-beat variability causes image artifacts. Real-time (RT) SSFP and recent innovations in image reconstruction provides a new potential alternative, capable of acquiring images without averaging over multiple heart beats. However, analysis of cardiac function from this image data can be complex, requiring retrospective analysis of function over multiple cardiac cycles and slices. We propose a deep learning regression method to facilitate cardiac phase detection, leveraging synthetic training approach from historical cine SSFP image data, and evaluate the effectiveness of this approach for detecting cardiac phase on RT SSFP images, manually labeled by expert readers. This combined approach using RT SSFP may have multiple potential advantages over traditional cine SSFP for evaluating cardiac function in patients with arrhythmia or difficulty tolerating long breath holds.

**Keywords:** Real-time (RT) steady-state free-precession (SSFP), data synthesizing, cardiac phase regression.

## 1. Introduction

Cine steady-state free-precession (SSFP) serves as the backbone of cardiovascular magnetic resonance imaging, and enables the quantitative assessment of left ventricular (LV) structure and function. Cine SSFP however, requires retrospective cardiac-gating with an electrocardiogram (ECG) to be recorded over multiple heartbeats and breath-holds (Wang et al., 2021). Because MRI signals are averaged over multiple RR-intervals using the ECG, beat-to-beat variations are obscured and image quality can be degraded by arrhythmia, in addition to patient motion or respiration. As an alternative, real-time (RT) SSFP, performed without ECG-gating, has the potential to address these limitations. However, quantitative analysis of RT SSFP images is time-consuming, as it requires identification of cardiac phase over multiple beats. (Chen et al., 2021; Rehman et al., 2022). To tackle this, we propose a semi-supervised deep learning strategy to automate identification of end-diastolic (ED) and end-systolic (ES) image frames for each short-axis slice across the length of the left ventricle, and facilitate estimation of ventricular volume and function.

## 2. Methods

In this IRB-approved, HIPAA-compliant study, we trained a convolutional neural network (CNN) to identify the ED and ES frames in successive cardiac cycles from RT SSFP images obtained in routine clinical care. Rather than using manual labels of cardiac phases on RT SSFP images, we instead elected to apply a synthetic training strategy (Masutani et al., 2020) to mimic RT SSFP images using historical short axis cine SSFP images from 241 cardiac MRI exams, previously labeled with cardiac phases. Since cine SSFP images are acquired with higher spatial and temporal resolution than RT SSFP, images were spatially and temporally downsampled to simulate real-time acquisitions. To provide an estimator of proximity to ED and ES phases, we applied a temporal Gaussian convolution ($\sigma = 75ms$) to ground truth ED and ES cardiac phase landmarks. The Gaussian-convolved proximity estimates were then used as labels for regression.

**Dataset** The data used for training included images from 241 cardiac MRIs, specifically short-axis cine SSFP images with temporal resolution ranging from $37 \pm 12.5ms$ and $256 \times 256$ spatial resolution. Data were split at a patient-level $80\% - 10\% - 10\%$ into training, validation, and test sets. In addition, the proposed CNN was evaluated using an independent set of 8 SAX RT SSFP acquisitions, obtained from a separate cohort of patients, which were annotated for ED and ES cardiac phases, by one of two physicians-in-training (A.M. and H.S.B), supervised by a board-certified radiologist with over 10 years of experience in cardiac MRI. RT SSFP images were obtained with $155.375 \pm 19.583ms$ temporal resolution and $512 \times 512$ spatial resolution.

**Model Development** We used Xception, pre-trained with ImageNet with 3-channel input (3 temporally successive frames) and a modified 2-channel output for ED and ES where each channel implied a $3 \times 1$ vector of ED and ES proximity estimation. Mean absolute error loss was used to regress the ED and ES proximity values according to the Gaussian ground truth around the ED/ES frames. Model was trained for 120 epochs using synthetic dataset and a batch size of 48, where augmention in form of random temporal resolution ($\delta t \in [120 - 240ms]$), starting point ($t_0$), and zoom-out took place during the training to simulate RT images with varying temporal resolution and larger filed of view. The best model (with lowest validation loss) was used to infer ED and ES from synthtetic test set. Results were averaged over time windows of 3, strides of 1 along the temporal axis. Thresholded (at 0.45), the second derivative of averaged Ed/ES predictions signified the elected ED/ES frames.

## 3. Results

Comparing the predictions to ground truth during the inference on the synthetic test set, results in terms of accuracy, and recall of ED, ES frame detection are provided using a maximum of 0-frame or 1-frame leniency (Table 1). Later, algorithm performance was assessed against the recorded ECG of the RT SAX SSFP images. Figure 1 depicts an example case with results for 3 ventricular slices along temporal resolution which confirms the efficiency of our proposed method to potentially skip the EKG gating, and to be able to extract R-R intervals from RT images and use the resulted synthetic cine SSFP images to evaluate cardiac function in form of a distribution for each cardiac measure rather than a scalar value.

Table 1: Results on synthetic test set

|  | 0-frame (0ms) | 1-frame $(120 - 240ms)$ |
| --- | --- | --- |
| Accuracy (ED) | 0.905 | 0.960 |
| Recall (ED) | 0.824 | 1.000 |
| Accuracy (ES) | 0.877 | 0.957 |
| Recall (ES) | 0.739 | 0.984 |

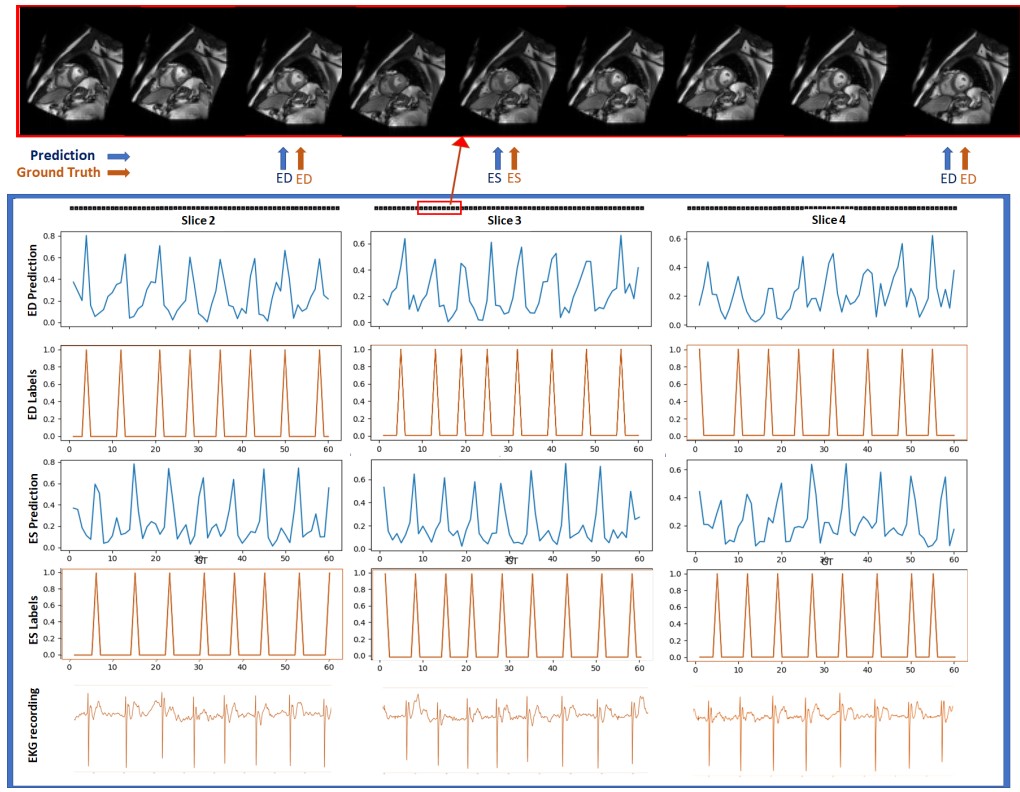

Figure 1: On the top, an exemplar time series of image frames from RT SSFP are shown along with image labels marking the ground truth annotations and algorithm inference. On the bottom, results of the deep learning regression algorithm are shown for 3 adjacent ventricular slices, along with ECG signals that were recorded concurrently to serve as an additional ground truth reference point.

## Acknowledgments

The authors would like to acknowledge GE healthcare for their support.

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
