# OpenReview forum: "Deep Learning Regression of Cardiac Phase on Real-Time MRI"
_MIDL.io/2023/Short_Paper_Track — MIDL 2023 Short paper track Poster_

### Official Review · Reviewer_9QDx · 2023-04-21
**This paper proposed a deep learning regression model to detect cardiac phase (end-systolic and end-diastolic), by using synthetic training data from historical cine SSFP image data and applying it on real SSFP data.**

**Rating:** 5
**Confidence:** 4

**Review:**

**Pros:**

- The paper is written clearly and is easy to follow.
- The idea of using synthetic training data to detect the cardiac phase cycle is interesting and addresses the problem of lack of data annotation.
- Real-time SSFP imaging enables continuous acquisition of images without the need for multiple heartbeats and breath-holds, as required in cine SSFP.
- The model was trained and tested on 241 cardiac MRI exams.

**Cons:**

- The novelty and technical contribution of this approach are limited.
- There is a lack of related work and performance comparison of the proposed method with existing cardiac phase detection approaches on cine MRI.
- It is not clear why the authors selected the Xception model for the regression task. Is there any specific reason for it?
- The authors referred to their approach as a semi-supervised deep learning strategy; however, it is unclear how this approach can be semi-supervised when the model was trained in a fully supervised manner.
- The authors mentioned in the paper that "The Gaussian-convolved proximity estimates were then used as labels for regression." It is not clear why proximity was used. Could the authors provide a rationale for this choice?
- Table 1 shows that the recall/sensitivity of ES is lower than ED. Can the authors comment on this difference?
- Based on Figure 1, there are some small spikes in between for both ES and ED predictions. Will these spikes have any impact on the clinical evaluation of these predictions?
- The plots need captions for axes to indicate what the x and y axes represent.

---

### Official Review · Reviewer_n2zi · 2023-04-24

**Rating:** 6
**Confidence:** 4

**Review:**

Disclaimer: I am not an expert in cardiac imaging and I may have missed previous literature that may have accomplished similar results.

Summary:

This work investigates whether ED and ES phases can be predicted by a neural network, when it processes Real Time (RT) Steady-State Free-Precession (SSFP), as this would be useful for image analysis. For this purpose, the authors use a previously acquired Cine SSFP that had ED and ES annotated. The Cine SSFP is of higher spatial and temporal resolution, and therefore it was subsampled to create a "synthetic" RT-looking SSFP, on which they trained a CNN to predict the ED and ES phases by regressing the "distance" of a frame from the ED/ES frames. The model was then evaluated on a held-out test set of the synthetic RT-looking SSFP, but also on real RT SSFP data, where experts had annotated ED and ES. The work reports quite high accuracy (90% ED, 88% ES for for 0-frame tolerance, and 96% for each for 1-frame tolerance).

Strength:
- Motivation for the application seems clear.
- The data is well written and the method well described
- Results seem promising

Weaknesses:
- There is very limited literature review, and the work does not state if this problem (regress/detect ED/ES frames in RT SSFP) has been previously tackled. As a result, it does not position itself within the literature.
- Limited technical contribution, if any (unclear as per above). The method applies a standard ResNet. If there is technical contribution, that could be in the way it approaches the problem (regression of the distance from a frame) but this is totally unclear as the work does not discuss any related work, and does not state if the problem has been approached before. My quick search did not return related work, but I may be missing something.
- The experimental analysis does not provide evidence and insights about "how" good are the results. There are no baselines to compare against. There is no analysis on whether this 90%, or 96%, are "good enough" for the method to be actually "useful" in some way. In future work, I would advise the authors to provide baselines, so that insights can be drawn about the quality of the method, and try to evaluate on whether performance is clinically useful, since the focus of the method is in the application and not technical contributions.